# CHIMERA: COMPOSITIONAL IMAGE GENERATION USING PART-BASED CONCEPTING

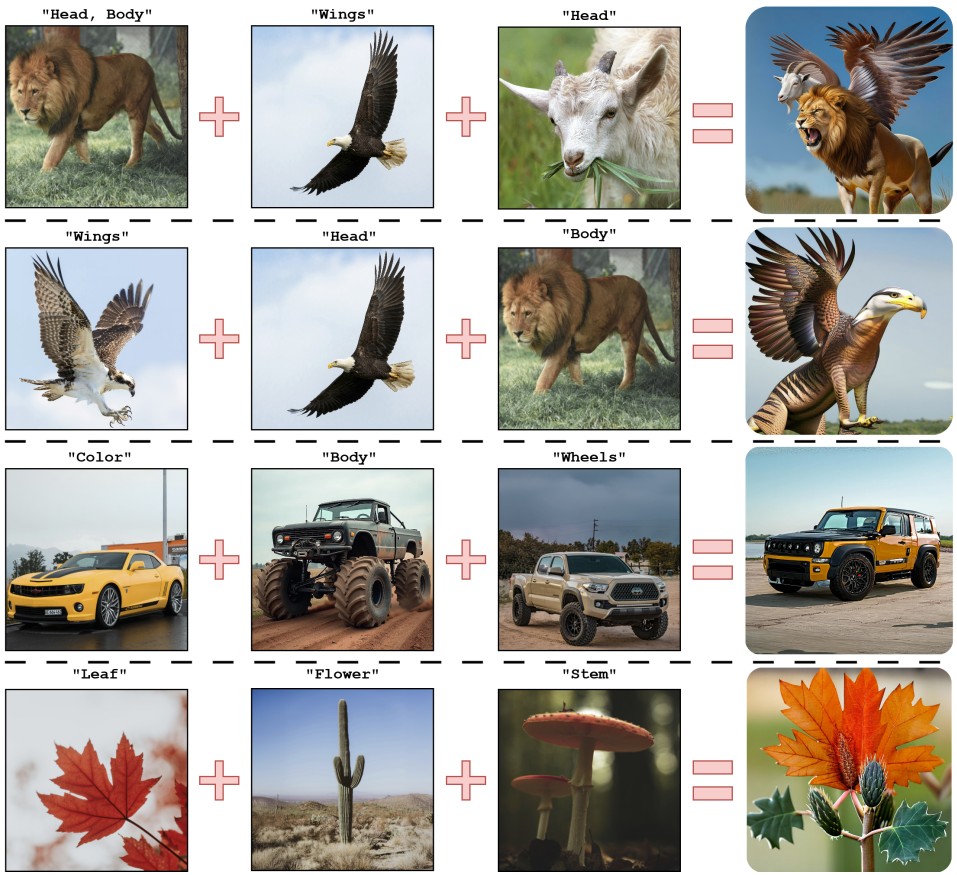

Figure 1: Part-aware composition with Chimera: the model takes input images along with their specified part prompts (e.g., head of a lion, body of a horse) and generates a new entity that combines these parts into a coherent output.

## ABSTRACT

Personalized image generative models are highly proficient at synthesizing images from text or a single image, yet they lack explicit control for composing objects from specific parts of multiple source images without user specified masks or annotations. To address this, we introduce Chimera, a personalized image generation model that generates novel objects by combining specified parts from different source images according to textual instructions. To train our model, we first construct a dataset from a taxonomy built on 464 unique ⟨part, subject⟩ pairs, which we term semantic atoms. From this, we generate 37k prompts and synthesize the corresponding images with a high-fidelity text-to-image model. We train a custom diffusion prior model with part-conditional guidance, which steers the image-conditioning features to enforce both semantic identity and spatial layout. We also introduce an objective metric *PartEval* to assess the fidelity and compositional accuracy of generation pipelines. Human evaluations and our proposed metric show that Chimera outperforms other baselines by 14% in part alignment and compositional accuracy and 21% in visual quality.

# 1 INTRODUCTION

Humans instinctively understand the world through parts [2; 9]: a scientist decomposes a system into modules, an artist sketches a figure starting from its head and limbs, and in daily life we describe objects in terms of distinctive components ("a car with big wheels," "a bird with a long beak"). This part-based perception enables flexible reasoning, creativity, and generalization.

By contrast, most visual recognition and generative models are trained on whole objects or entire concepts, without an explicit notion of their constituent parts. As a result, these models often learn entangled global features and provide only coarse control—users can modify style or add objects [3; 7], but cannot directly manipulate specific parts (e.g., "change just the head of the bird). This gap highlights a key mismatch: while humans naturally reason in terms of parts, models operate at the level of indivisible wholes. In this work, we explore how image synthesis can benefit from the same part-aware principles that guide human understanding; we move beyond monolithic representations to a method that composes and personalizes objects explicitly at the part-level.

Early personalization methods such as DreamBooth [26], Textual Inversion [4], and Custom Diffusion [10] adapt a generative model to capture novel concepts by associating them with unique text embeddings or fine-tuned weights. However, these methods **1)** operate at the level of entire concepts rather than object parts, and **2)** require 3 to 5 reference images per concept to achieve reasonable fidelity. As a result, they are restricted in their ability to compose or disentangle fine-grained part-level attributes, and struggle to generalize beyond whole-concept personalization.

More recent work has begun to explore part-oriented generation. For instance, Piece-It-Together (PiT) [25], PartCraft [15], and PartComposer [11] all introduce mechanisms to compose images from object parts rather than whole concepts. However, these approaches **1)** require explicit user-provided masks to isolate parts. **2)** generalize poorly. PartCraft, for example, is constrained to specific categories such as birds and dogs, while PiT primarily targets toy creatures and product-like objects. As a result, these methods fall short of supporting broad and generalizable part-based image synthesis and have inconsistencies when integrating multiple heterogeneous components.

To address these limitations, we propose **Chimera**, a personalized image generation model that enables the creation of novel hybrid objects by combining parts from different input images using only the images and their corresponding text prompts. Unlike prior methods, Chimera does not require users to provide explicit masks or annotations; only whole images with prompts are sufficient. The model maintains consistency and quality even as the complexity of the creation increases, producing coherent results for 2-part, 3-part, and 4-part compositions (see Figure 2). To support broad generalization, Chimera is trained on an extensive taxonomy of everyday objects spanning animals, vehicles, and artifacts, allowing it to generate diverse part-level combinations beyond narrow

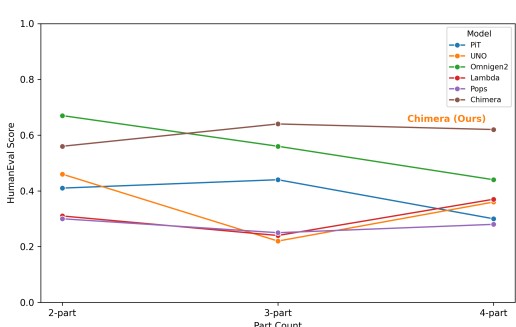

Figure 2: Scores assigned by human annotators for object generations of increasing complexity. For example, 2-part refers to generating objects using two provided image-text pairs.

domains. To construct training data, we develop a synthetic pipeline that leverages HiDream-Full [8] for prompt-specific image generation and Florence [31] for annotation. We then fine-tune the PiT model on this data and design a composite pipeline that combines Grounding DINO [22] and PiT [25]. Importantly, Chimera does not require explicit part masks or annotations: Users simply provide whole input images and text prompts, and the system automatically generates hybrid objects. In summary, our contributions are as follows:

- Chimera is a personalized image generative model with part-level conditioning. Rather than combining whole concepts, it combines different parts from single images to generate hybrid objects from multiple inputs.

- The model does not require masks or part annotations; it only requires whole images and accompanying text prompts as inputs.

- Evaluation through human studies and the new proposed PartEval metric demonstrates that Chimera preserves quality consistently across 2, 3, and 4-part generations, in contrast to prior models whose fidelity declines as complexity increases. We report a **14%** increase in compositional accuracy and part alignment across all categories compared to the next-best baselines.

## 2 RELATED WORK

**Personalization methods.** Personalization methods extend the distribution of a pretrained model to include a specific concept provided by a set of images. Most approaches achieve this either by learning specialized text embeddings to represent the concept [4] or by fine-tuning layers of the model itself [10; 26] . These methods learn a single concept from one or several images, and can then incorporate the learned concept in new images generated according to text prompts. In contrast, we focus on multi-concept disentangled personalization and composition, i.e. extracting several concepts from a single image, and composing them with concepts learned from other images.

**Compositional Multi-Concept Blending** Several recent works aimed to learn multiple concepts from a single or few images, enabling these concepts to be reused in novel compositions. Break-a-Scene [1] employs Dreambooth finetuning [26] while relying on user-provided spatial masks to isolate the concepts in the image. The dependence on spatial masks poses a significant limitation, as does not allow to disentangle e.g. the appearance of an object from its pose. Other methods, like Inspiration Tree [27] and ConceptExpress [5], extract multiple concepts by jointly learning several tokens from a single image, each representing a different concept. However, the disentanglement achieved by these methods is unpredictable, offering no control over which aspects of the image are separated into individual concepts.

Recent advances in compositional image synthesis have broadened the capabilities of diffusion models to fuse and manipulate multiple high-level concepts. Some methods refine cross-attention maps to steer semantic emphasis during generation [3; 7]. Others leverage vision-language priors to constrain how concepts are realized and guide creative variation [12; 24]. A further line of work benchmarks the difficulty of composing multiple concepts and explores embedding-level fusion strategies for seamless blending [14; 16; 30]. However, none of these approaches provides explicit control at the part-level, resulting in misaligned or inconsistent synthesis when recombining finer-grained components.

**Part-Based Representation Learning.** A rich body of work on part-based representation learning has produced curated taxonomies and large-scale annotated datasets for modeling object structure. These supervised resources provide semantic vocabularies and dense part masks in diverse categories [6; 19; 21]. Complementary unsupervised methods have focused on discovering and disentangling latent part-like factors directly from visual data, reducing the dependence on manual annotation [13; 25]. Although these efforts establish the atomic units necessary for compositional modeling, they remain untapped in diffusion-based generators, which currently lack mechanisms to incorporate such part-level priors during synthesis.

Chimera builds on the Piece-it-Together (PiT) [25] model, but achieves significantly greater generalizability through an expansive taxonomy of ⟨part, subject⟩ elements. This taxonomy spans a wide range of categories, including everyday objects, vehicles, and animals, which provides the model with richer and more diverse training. Unlike prior methods, users are not required to supply isolated part images: instead, they can simply provide whole input images along with a text prompt. Moreover, Chimera imposes no restriction on the number of input images, enabling seamless integration of multiple parts and subjects within a single generated output.

## 3 METHOD

This section introduces the synthetic data creation framework to train the model that can do part-conditioned generation. First, we will detail the preliminaries about diffusion-based prior . Next, we will present our synthetic data creation pipeline. Finally, we will share our training details.

| Model | Learns new tokens? | Finetuning strategy | Input images required per concept | Masks required? |
|---|---|---|---|---|
| PiT [25] | ✗ | Prior model† | 1 | ✓ |
| UNO [29] | ✗ | Full model | 1 | ✗ |
| $\lambda$-eclipse [17] | ✗ | Prior model† | 1 | ✗ |
| OmniGen2 [28] | ✗ | Full model | 1 | ✗ |
| pOps [23] | ✗ | Prior model† | 1 | ✗ |
| Textual Inversion [4] | ✓ | Learnable token only | 1–5 | ✗ |
| DreamBooth [26] | ✓ | LoRA | 3–5 | ✗ |
| Custom Diffusion [10] | ✓ | K/V layers | 5–10 | ✗ |
| PartCraft [15] | ✓ | LoRA | 1 | ✗ |
| PartComposer [11] | ✓ | LoRA | 1 | ✓ |
| **Chimera (Ours)** | ✗ | Prior model† | 1 | ✗ |

Table 1: Comparison of personalized image synthesis methods. The table shows whether each method learns new tokens, what kind of fine-tuning it uses, how many input images are needed per concept, and whether users must provide masks.

†*Prior model refers to a relatively smaller pretrained model first introduced in the DALL-E paper [20].*

## 3.1 PRELIMINARIES

### 3.1.1 DIFFUSION PRIOR

Ramesh et al. [20] introduce the Diffusion Prior model, tasked with mapping an input text embedding to a corresponding image embedding in the CLIP [18] embedding space. This image embedding is then used to condition the generative model to generate the corresponding image. This mechanism allows them to not only use existing image embeddings as a condition but also generate such inputs using a separate generative process.

Diffusion models are typically trained with a conditioning vector, $c$, directly derived from a given text prompt, $y$. Ramesh et al. [20] introduce a two-stage approach that decomposes the text-to-image generation process into two steps.

First, a diffusion model is trained to generate an image conditioned on an image embedding, $c$, using the standard diffusion loss:

$$L_{\text{diffusion}} = \mathbb{E}_{z,y,\epsilon,t} \left[ ||\epsilon - \epsilon_\theta(z_t, t, c)||^2 \right]. \tag{1}$$

Here, the denoising network, $\epsilon_\theta$, learns to remove noise $\epsilon$ added to the latent code $z_t$ at time step $t$, given the conditioning image embedding $c$.

Next, the Diffusion Prior model, $P_\theta$, is trained to recover a clean image embedding, $e$, from its noisy version, $e_t$, conditioned on the corresponding text prompt, $y$. The objective function is given by:

$$L_{\text{prior}} = \mathbb{E}_{e,y,t} \left[ ||e - P_\theta(e_t, t, y)||^2 \right]. \tag{2}$$

Once both models are trained separately, they are combined into a full text-to-image generation pipeline. In this work, we extend the Diffusion Prior beyond its conventional role of predicting image embeddings from text. Instead, we adapt it to operate over multiple image parts, enabling finer-grained control over the synthesis process.

### 3.1.2 IP-ADAPTER

IP-Adapter [32] is a lightweight image prompt adaptation method with the decoupled cross-attention strategy for existing text-to-image diffusion models. It helps to steer image generation using image embeddings along with the standard text embeddings.

### 3.1.3 PIECE-IT-TOGETHER

Piece-It-Together [25] builds the IP-Adapter+ embedding space , on which we train IP- Prior, a lightweight flow-matching model that synthesizes coherent compositions based on domain-specific priors, en- abling diverse and context-aware generations.

## 3.2 DATA GENERATION

We used the open-source HiDream-I1 Full text-to-image model (**HiDream-ai/HiDream-I1-Full**) [8] provided on Hugging Face to compile a large-scale, semantically varied dataset of hybrid animal pictures in order to train and rigorously assess our framework. We chose HiDream-I1 Full because it has proven to be able to render intricate anatomy and maintain style during several runs, allowing for accurate part-level control in a synthetic environment.

### 3.2.1 PART TAXONOMY DESIGN

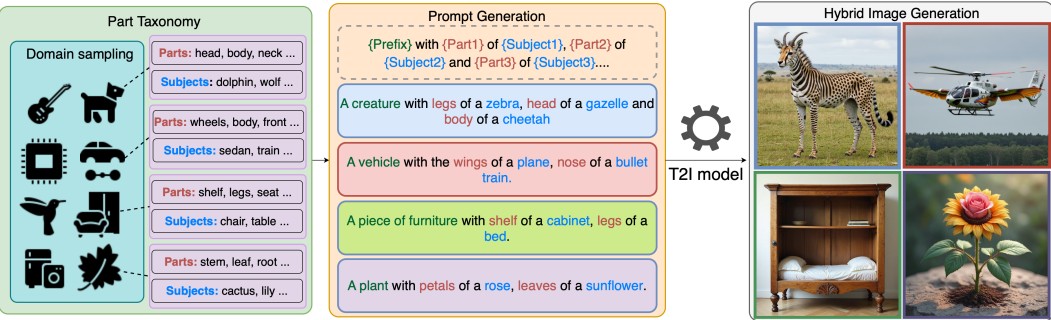

Figure 3: Data creation pipeline: From a multi-domain part taxonomy, we sample 2-4 subject-part pairs and generate a prompt, render hybrids with HiDream-I1 Full, and collect images with ground-truth part masks for training and evaluation.

Throughout the paper we use *part* to denote a visually localized, semantically meaningful component (e.g., *wheels*, *screen*, *stem*), and *subject* to denote the object category or species that supplies that part (e.g., *sports car*, *laptop*, *rose*). A semantic atom ⟨part, subject⟩ therefore specifies both *what* the component is and *whose* component it is—for example, ⟨tail, lion⟩ or ⟨keyboard, laptop⟩. As illustrated in Figure 3, our taxonomy covers 6 semantic domains: *creature, vehicle, furniture, plant, electronics*, and *instrument*.

With 8 parts per domain and each 6 to 19 subjects, the taxonomy yields 464 unique ⟨part, subject⟩ semantic atoms. Each atom maps to a coherent, visually localized region (e.g., *wheels* on a *sports car*), simplifying mask alignment and evaluation. Ambiguous surface attributes such as color or texture are excluded to preserve part-level clarity.

Hybrid prompts are generated by sampling 2 to 4 distinct atoms, optionally mixing domains, to balance linguistic clarity with compositional richness. Capping at 4 atoms keeps prompts concise while challenging the diffusion model to assemble multiple heterogeneous components. A category-specific prefix is prepended (e.g., `A creature`, `A vehicle`), and the remaining template is:

```
{Prefix} with {Part_1} of a {Subject_1}, {Part_2} of a
{Subject_2}, and {Part_3} of a {Subject_3}.
```

This strategy supports millions of possible hybrids while ensuring clear, part-to-image correspondence.

### 3.2.2 IMAGE SYNTHESIS WITH HIDREAM-I1 FULL

Every hybrid prompt is rendered into an image using HiDream-I1 Full [8], an open-source Diffusion Transformer based image generation model released on HuggingFace that excels at anatomical detail and clean object boundaries. All images are generated at $1024 \times 1024$ px using 50 denoising steps, a guidance scale of 5.0, and a scheduler shift of 3.0. A deterministic seed derived from the prompt string guarantees reproducibility while preserving diversity across the 37k-prompt corpus. No additional post-processing or filtering is applied; the resulting 37k images constitute the raw corpus for training our part-conditioned diffusion prior described in the next stage. Qualitative inspection confirms that HiDream-I1-Full faithfully renders the specified parts while plausibly hallucinating unspecified regions, as illustrated in Figure 3.

## 3.3 MODEL PIPELINE

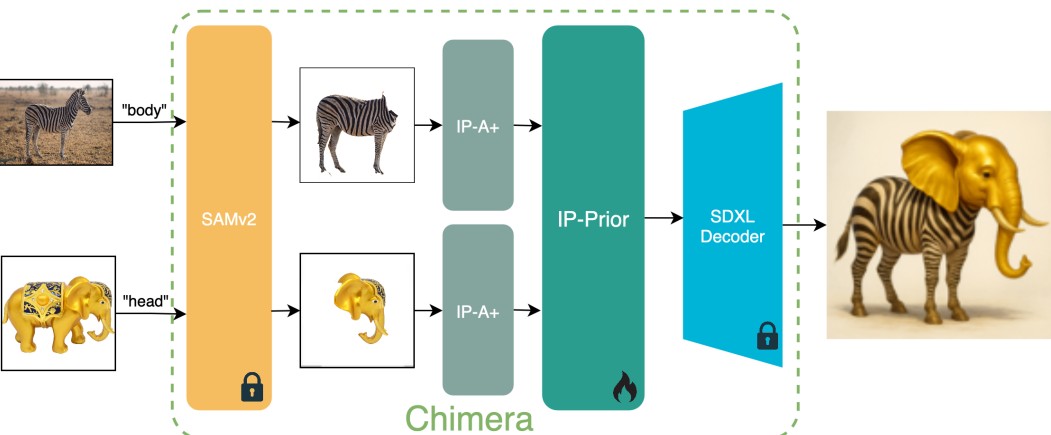

Figure 4: An overview of our generation pipeline. User inputs are processed by Grounded-Segment-Anything (SAMv2) to produce segmented images, which are then encoded into the IP+ embedding space. This embedding conditions our finetuned PiT model, acting as a rectified flow prior, to generate a target latent that is subsequently rendered into the final image by the SDXL decoder.

We build upon the PiT model, finetuning the rectified flow prior on the dataset we created. The input images from the user along with the text prompts are passed to the Grounded-Segment-Anything model to produce cropped images containing the desired parts. These images are then converted into image embeddings in the IP+ space, which is used as conditioning by the prior model to generate the latent target embedding. The SDXL decoder then operates upon the target embedding in the IP+ embedding space to create the output image.

## 3.4 TRAINING DETAILS

The decoder is frozen, meanwhile we train just the prior on the source and target embeddings from the generated dataset for 500k training steps on a single A100 GPU.

## 4 PARTEVAL

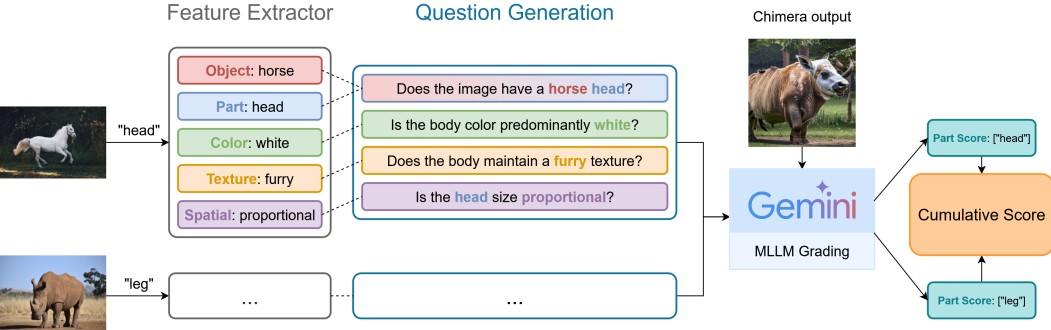

Figure 5: Our proposed MLLM-based evaluation pipeline. The process consists of three stages: (1) extracting structured features (e.g., color, texture) from a reference image, (2) automatically generating questions from these features, and (3) using Gemini-Flash to grade the generated image's faithfulness by answering the questions.

To evaluate the performance of our part-influenced generation, we propose a new metric that leverages Multimodal Large Language Models (MLLMs). Our evaluation pipeline is composed of three distinct stages:

1. **Feature Extraction**: The first stage acts as a feature extractor. Given a reference image and a part-specific prompt, this module extracts a structured description detailing the part's key attributes: **Object**, **Part**, **Color**, **Texture**, and **Spatial Relation**.

2. **Question Generation**: In the second stage, the structured features extracted in the previous step are automatically converted into a set of targeted evaluation questions.

3. **Grading**: The final stage involves grading the generated image against the questions from the previous step. We use Gemini-Flash to answer each question based on the image's content. A response is awarded 1 point if it confirms the specified attribute is faithfully represented, and 0 points otherwise. The sum of these points for a single image constitutes its partial score.

Finally, we compute the comprehensive final score by averaging the partial scores across all evaluated images. This average is then normalized to a scale of 0 to 1.

## 5 EXPERIMENTS

### 5.1 QUALITATIVE COMPARISONS

In Figure 6, we present a range of 2-part compositions in the animal and vehicle categories generated by various methodologies including PiT, pOps, OmniGen2, UNO, $\lambda$-eclipse and Chimera . Chimera demonstrates great proficiency in composition while ensuring part alignment. In contrast, the baselines often overemphasize input images; they are capable of preserving one of the input images but fail to properly combine multiple parts from different images. We also present results on 3-part and 4-part compositions in the appendix, which depict the consistency of our method with increasing complexity.

### 5.2 QUANTITATIVE EVALUATIONS

We evaluate image quality using FID and KID, which measure the similarity between generated samples (across all possible part combinations) and reference datasets for the corresponding categories. The reference datasets are generated with HiDream-Full, the same model used for training data synthesis. Results are reported in Table 2.

In addition, we introduce compositional accuracy, which measures how well a model preserves the identity of input parts and reproduces them in the generated output. To quantify this, we use the PartEval metric, scoring 200 generated samples from both the animal and vehicle categories on a 0-10 scale. Table 3 presents the results, showing that Chimera consistently achieves higher compositional accuracy compared to baseline models. Finally, to assess generalizability, we conduct human evaluations. Participants rate generated images on a 0-10 scale, and scores are normalized to the range [0,1]. For 2-part and 3-part evaluations, we average ratings from 150 respondents over 25 samples per model. For 4-part evaluations, we collect ratings from 30 respondents across 10 generated samples. The **Bold** and Underline values in the tables indicate the best and second-best score in criteria respectively.

Analysis suggests that Chimera achieves noticeably better visual quality, as reflected by its lower FID and KID scores. Importantly, its compositional accuracy remains consistent even as the number of parts increases, with OmniGen2 emerging as the closest competitor in terms of both quality and part alignment. The baseline models perform reasonably well on simple 2-part compositions, but their performance degrades noticeably as the compositional complexity increases. Both the PartEval and human evaluation results follow a similar trend across categories and levels of complexity. However, we observe that PartEval generally assigns higher scores than human evaluators, indicating a sys-

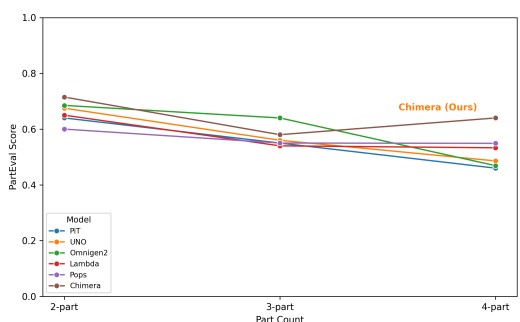

Figure 7: Scores assigned by PartEval metric for object generations of increasing complexity.

tematic optimism in automated part-based assessment compared to subjective human judgment. This is further illustrated by the differences between Figure 2 and Figure 7. In Figure 7, the lines for different models cluster closely together, suggesting that the PartEval metric is less sensitive to

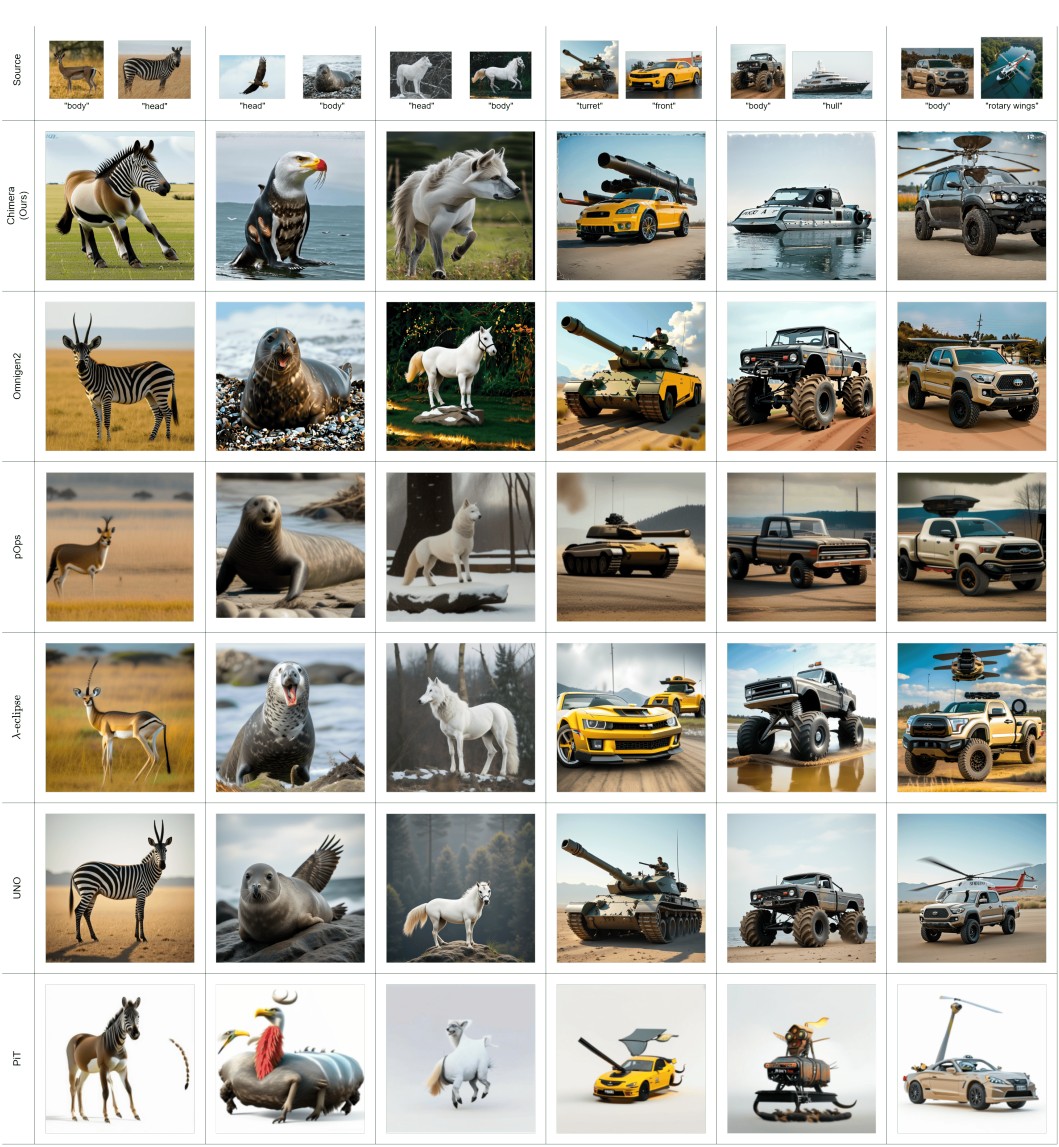

Figure 6: Qualitative comparisons for animals and vehicles.

qualitative differences across models. By contrast, Figure 2 shows more separation between methods, reflecting absolute human preferences more faithfully and highlighting cases where PartEval may overestimate alignment.

| Category | Metrics | PiT [25] | UNO [29] | OmniGen2 [28] | $\lambda$-eclipse [17] | pOps [23] | Chimera (Ours) |
|---|---|---|---|---|---|---|---|
| Animals (2 parts) | FID ↓ | 23.95 | 27.91 | 26.20 | 24.65 | 25.95 | **22.88** |
|  | KID ↓ | 0.031 ± 0.005 | 0.044 ± 0.006 | 0.039 ± 0.005 | 0.033 ± 0.004 | 0.034 ± 0.004 | **0.027** ± 0.004 |
| Animals (3 parts) | FID ↓ | 24.99 | 27.08 | 25.22 | 24.66 | 25.13 | **23.85** |
|  | KID ↓ | 0.038 ± 0.004 | 0.040 ± 0.005 | 0.035 ± 0.005 | 0.038 ± 0.005 | 0.029 ± 0.003 | **0.029** ± 0.004 |
| Vehicles (2 parts) | FID ↓ | 31.21 | 30.74 | 32.16 | 32.77 | 39.98 | **30.20** |
|  | KID ↓ | 0.063 ± 0.005 | **0.047** ± 0.004 | 0.094 ± 0.006 | 0.085 ± 0.007 | 0.118 ± 0.007 | 0.063 ± 0.006 |
| Vehicles (3 parts) | FID ↓ | 32.32 | 30.84 | 30.72 | 33.82 | 40.15 | **30.44** |
|  | KID ↓ | 0.074 ± 0.005 | **0.067** ± 0.006 | 0.087 ± 0.005 | 0.082 ± 0.006 | 0.102 ± 0.006 | 0.069 ± 0.005 |

Table 2: Generation quality comparison across Animals and Vehicles with different part counts. Lower is better for FID and KID.

| Category | Metrics | PiT [25] | UNO [29] | Omni-Gen2 [28] | $\lambda$-eclipse [17] | pOps [23] | Chimera (Ours) |
|---|---|---|---|---|---|---|---|
| Animals (2 parts) | PartEval ↑ | 0.63 | 0.66 | 0.66 | 0.61 | 0.57 | **0.71** |
|  | HumanEval ↑ | 0.52 | 0.56 | **0.57** | 0.30 | 0.30 | 0.52 |
| Animals (3 parts) | PartEval ↑ | 0.56 | 0.57 | 0.61 | 0.53 | **0.62** | 0.58 |
|  | HumanEval ↑ | 0.33 | 0.22 | 0.42 | 0.23 | 0.29 | **0.58** |
| Vehicles (2 parts) | PartEval ↑ | 0.65 | 0.69 | 0.71 | 0.71 | 0.64 | **0.73** |
|  | HumanEval ↑ | 0.33 | 0.35 | **0.76** | 0.31 | 0.31 | 0.61 |
| Vehicles (3 parts) | PartEval ↑ | 0.54 | 0.57 | **0.67** | 0.54 | 0.50 | 0.60 |
|  | HumanEval ↑ | 0.33 | 0.21 | 0.69 | 0.27 | 0.24 | **0.70** |

Table 3: Comparison across Animals and Vehicles using **PartEval** (automatic part-level fidelity metric) and **Human Evaluation**. Higher is better.

# 6 CONCLUSION

We introduce Chimera, a personalized image generation model that enables new visual generation capabilities through part-level composition. Finetuned on an extensive part-species taxonomy, Chimera is able to learn how to disentangle and compose fine-grained part-level concepts from single-image examples. Beyond academic benchmarks, such a method opens avenues for practical applications: character creation in gaming, where artists can flexibly mix features across creatures or avatars; object design in animation pipelines for VR/AR, where hybrid parts enable rapid prototyping of immersive assets; or industrial design, where novel variants can be generated by recombining functional components. More broadly, we argue that engraving part-based capabilities into generative models is key for frontier-scale training, ensuring models can reason over whole concepts aswell as their overall compositional structure. This is an essential step towards controllable and reliable multimodal model training to make them truly capable of the way humans see the world.

# 7 ETHICS STATEMENT

All datasets used in this work are either publicly available or synthetically generated, and no private or sensitive information is included. The proposed methodology is intended for research purposes in areas such as creative design, animation, and data augmentation and aligns with responsible AI research practices. We follow the ICLR Code of Ethics and affirm that this work raises no conflicts of interest, privacy risks, or violations of research integrity.

# 8 REPRODUCIBILITY STATEMENT

We will release the code in an anonymous GitHub repo linked here: `Chimera Repository`.

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

# A APPENDIX

## A.1 THE USE OF LARGE LANGUAGE MODELS (LLMS)

In the preparation of this work, Large Language Models (LLMs) were used as an assistive tool to refine the writing style, grammar, and readability of the manuscript. In addition, they were employed in the process of surveying related literature by helping identify and summarize relevant prior work. All conceptual contributions, technical methods, and experimental designs presented in this paper are original, and the role of LLMs was limited to language editing and literature discovery support.

## A.2 MORE QUALITATIVE COMPARISONS

We present more qualitative comparisons for 2, 3 and 4 part creations in Figure 8, 9, 10 and 11.

## A.3 COMPARATIVE PERFORMANCE USING PARTEVAL AND HUMAN EVALUATION

To provide a detailed analysis of our model's capabilities, Figure 12 presents a direct performance comparison between our method and other baselines. The comparison spans five challenging compositional categories: 2- and 3-part animals, 2- and 3-part vehicles, and 4-part creations.

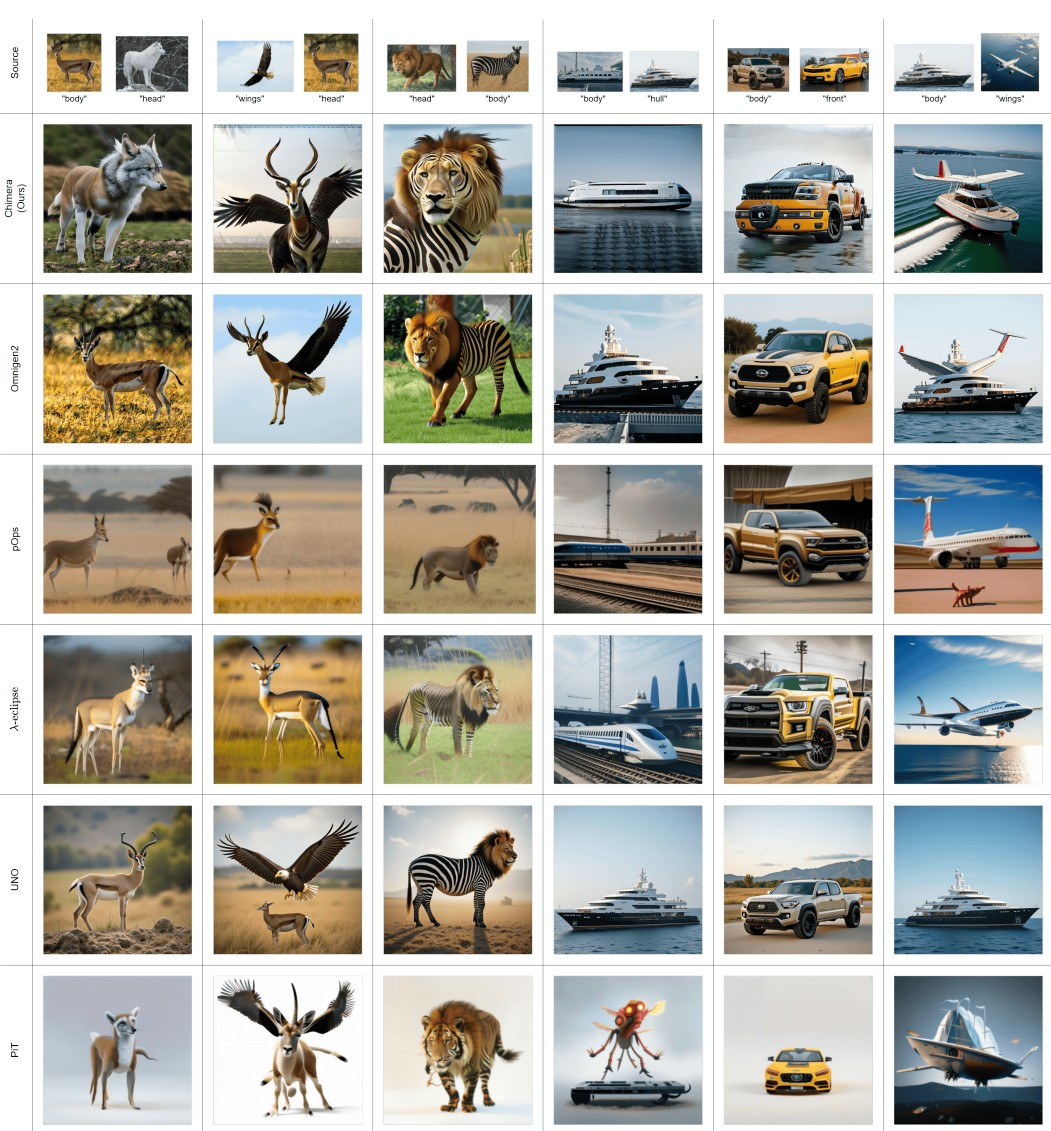

Figure 8: Qualitative evaluations for 2-part animals and vehicles.

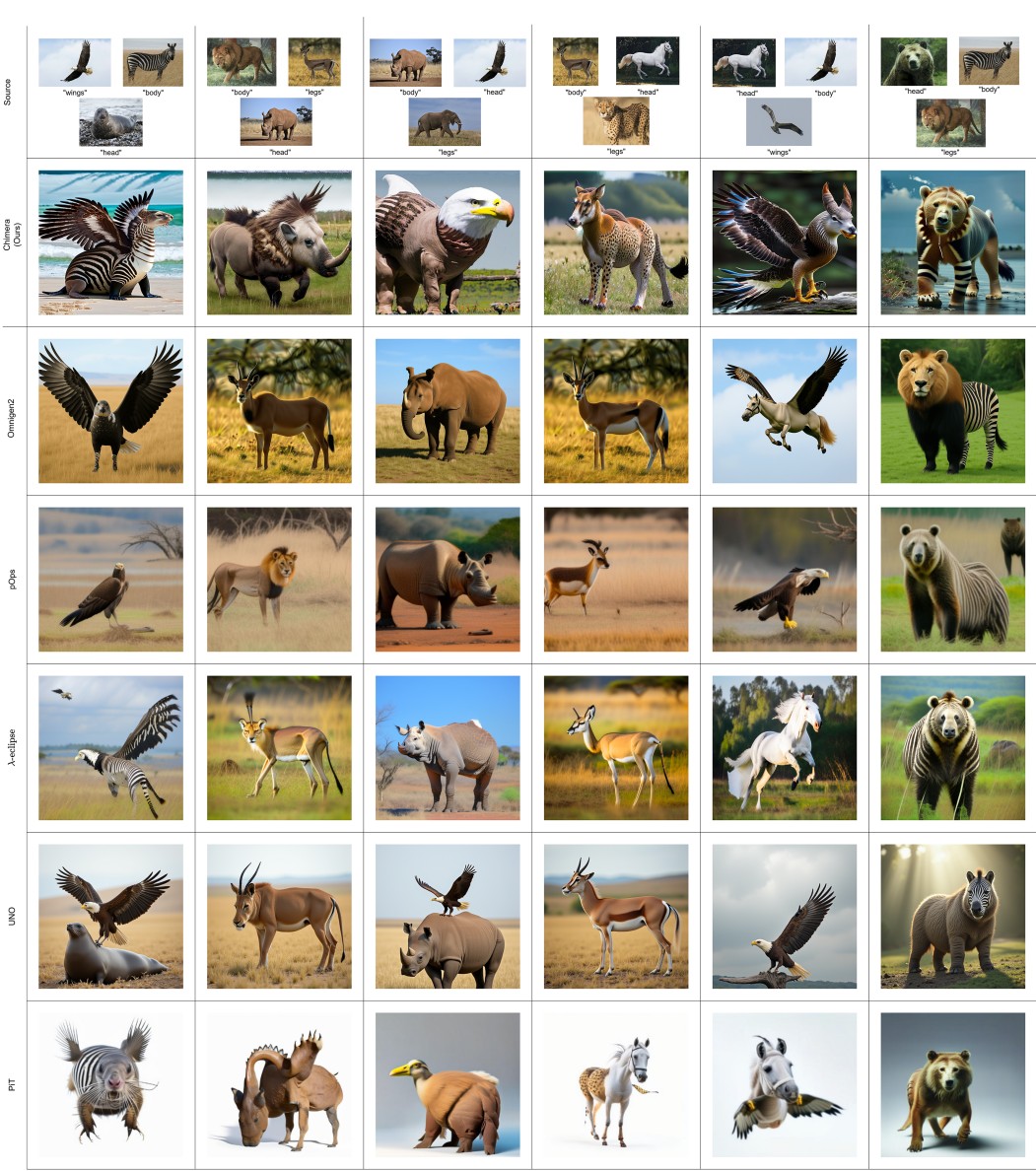

Figure 9: Qualitative evaluations for 3-part animals.

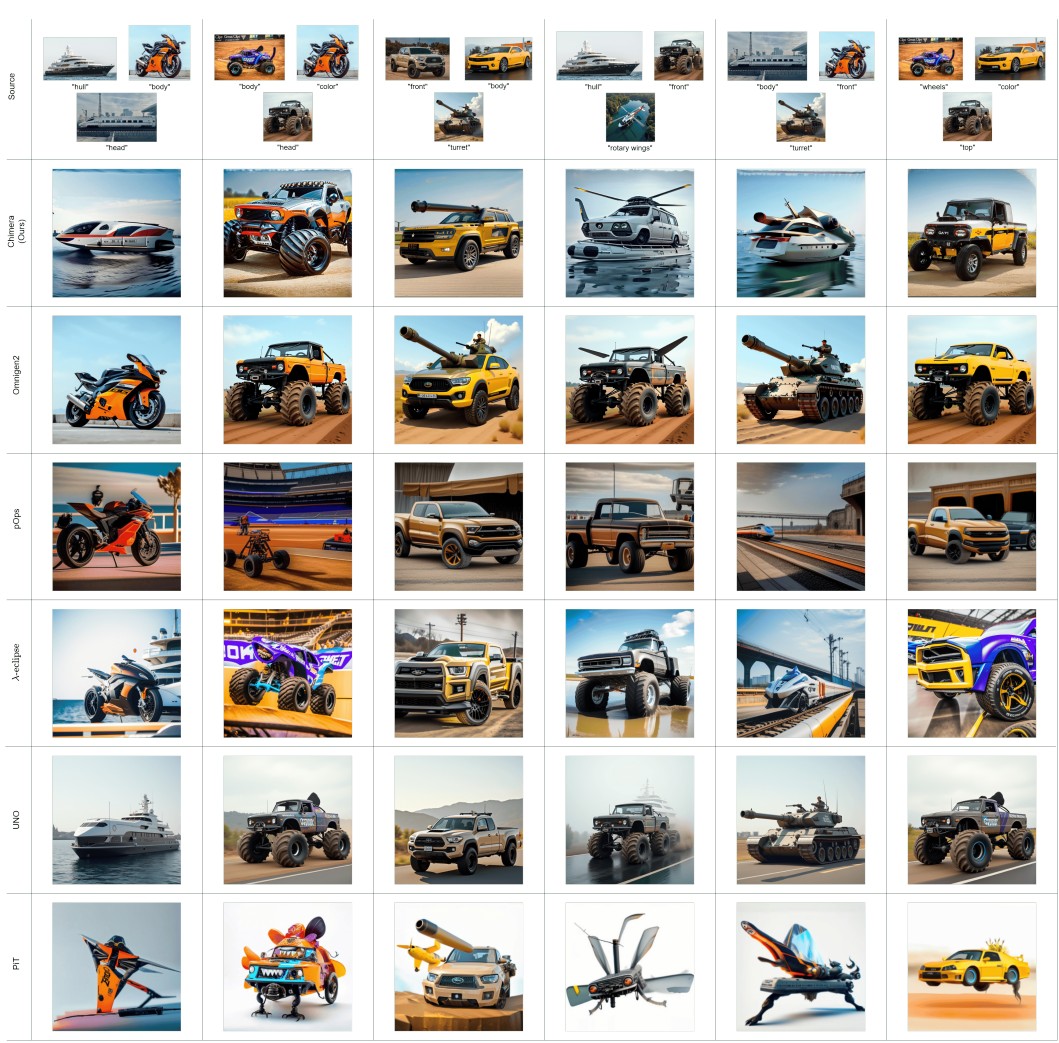

Figure 10: Qualitative evaluations for 3-part vehicles.

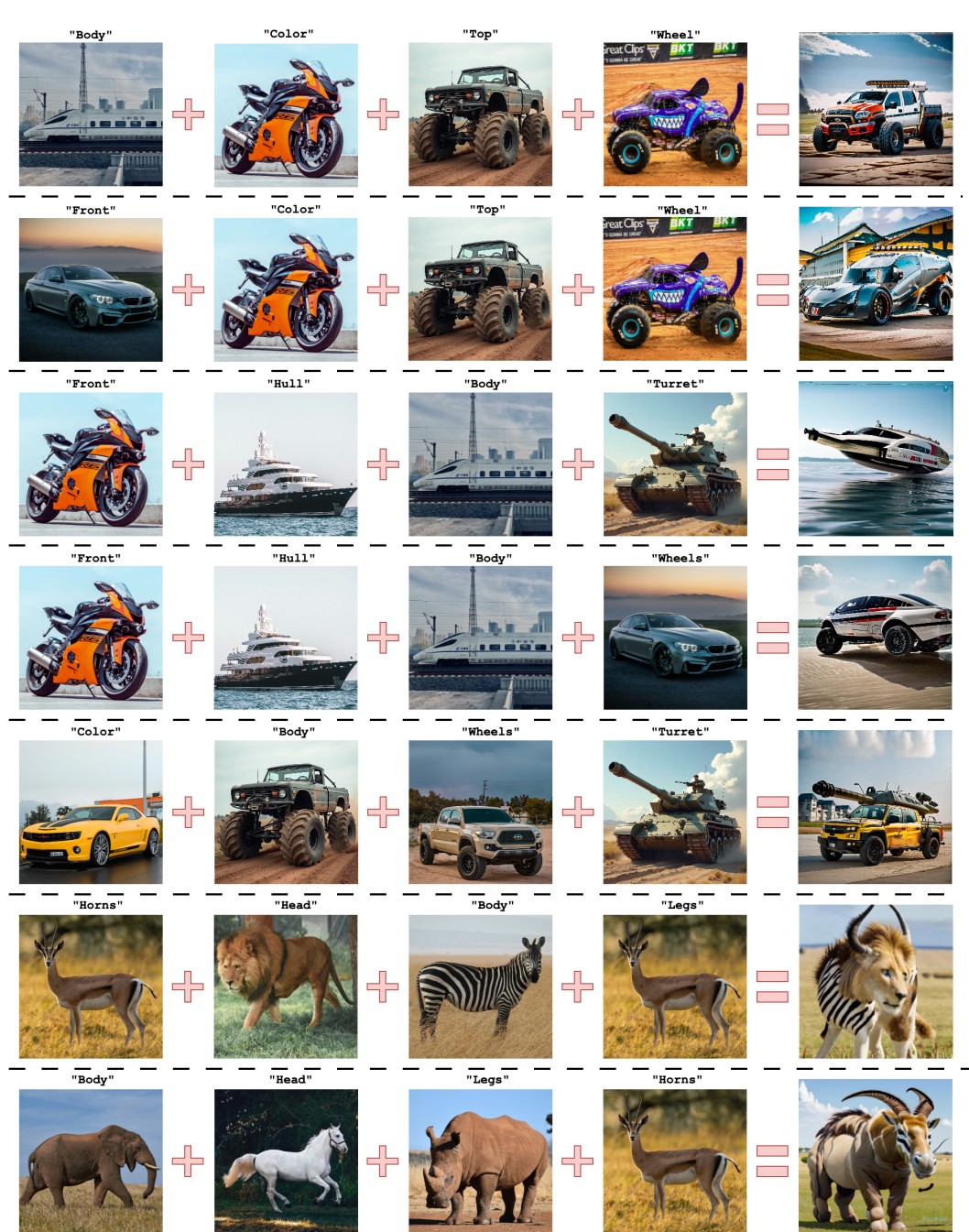

Figure 11: Some results with Chimera for 4-part compositions.

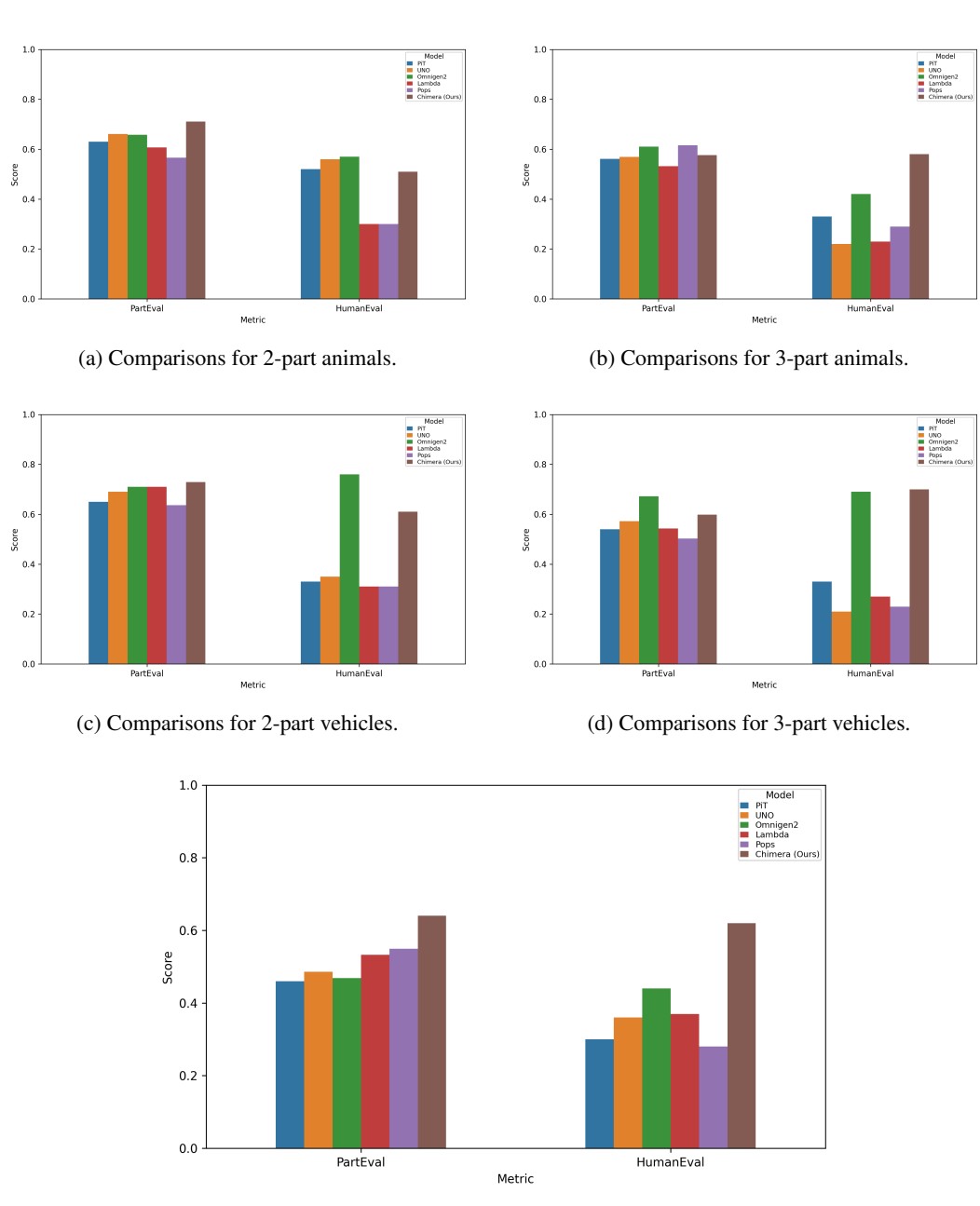

(a) Comparisons for 2-part animals.

(b) Comparisons for 3-part animals.

(c) Comparisons for 2-part vehicles.

(d) Comparisons for 3-part vehicles.

(e) Comparisons for 4-part creations.

Figure 12: PartEval and HumanEval comparisons across animals, vehicles, and multi-part creations.

