# OpenReview forum: "Chimera: Compositional Image Generation  using Part-based Concepting"
_ICLR.cc/2026/Conference — Submitted to ICLR 2026_

### Official Review · Reviewer_RB71 · 2025-10-26

**Soundness:** 2
**Presentation:** 3
**Contribution:** 1
**Rating:** 2
**Confidence:** 4

**Summary:**

This paper proposes Chimera, which is a framework for compositional generation based on parts. The authors propose a technique based on learning of IP-Adapters for each part, output of which is then composed to get the required image. The authors create a dataset for training of HiDream-11 model to generate the dataset.

**Strengths:**

Paper is clearly written and is understandable.

The problem is interesting to solve.

**Weaknesses:**

Results are Unnatural and contain artifacts: In Fig. 1, the results in rows 1,2, and 4 look quite unnatural; further, there are artifacts in the head region of Row 1.
 Novelty: I find the proposed approach to be very similar to the approach in Piece-It-Together. Hence, the novelty of the proposed method is unclear.

Soundness: I find that the dataset for learning itself contains artifacts (Fig. 1, showing the aeroplane and sunflower). So unclear to me how it can be used to train a model to do a part composition task with bad supervision?

The performance of the Omni-Gen2 and Chimera is very similar. Hence, the exact advantage of the method needs to be clarified.

**Questions:**

The part identity of the parts is not similar to the input images provided (Fig. 6). Could authors provide a reason for that?

Some recent works like [21, 11, 15] are not considered as baselines for comparison. Could the authors provide reasoning for that?

---

> ### Author Response · Authors · 2025-11-20
>
> Thank you for your time and your review of our work. Below, we address your questions and the weaknesses mentioned.
>
> **Novelty & advantages of proposed method:** While inspired by Piece-It-Together, our method significantly extends beyond its limited scope (cartoon/product domains) by introducing a broad taxonomy that generalizes to real-world categories. Our model maintains strong performance across increasing compositional complexity (2–4 parts) unlike other current approaches which fail to produce coherent creations at higher complexities. Additionally, we propose the new metric PartEval to assess compositional ability, offering a foundation for future research.
>
> **Recent works not considered as baseline:** The referenced methods rely on LoRA-based fine-tuning with a few specific concept images, making them unsuitable for true zero-shot generation. In contrast, our approach operates without such fine-tuning. Additionally, works like PartCraft and DreamCreature are limited to narrow domains (e.g., birds, dogs), reducing their generalizability to broader, real-world categories.

---

### Official Review · Reviewer_aNSa · 2025-10-31

**Soundness:** 2
**Presentation:** 3
**Contribution:** 2
**Rating:** 6
**Confidence:** 4

**Summary:**

The paper introduces Chimera, a pipeline for compositional image generation that aims to combine specified parts from different source images based on text prompts. The core methodology relies on three key components: (1) SAMv2 for identifying parts, (2) Diffusion Prior (IP-Prior) trained on a large, synthetic corpus of hybrid concepts, and (3) a decoder. Also, the authors propose a metric PartEval, using Gemini-Flash for evaluating semantic compositional quality of the generated images.

**Strengths:**

1. By intelligently integrating a strong segmentation model (SAMv2), the approach eliminates the primary usability bottleneck for previous part-based generation models.
2. Part-Eval provides a novel metrics that is essential for measuring the success of compositional tasks. This moves the field beyond general image quality metrics (like FID) which are insufficient for judging correct part-blending.
3. Its interesting to see how the proposed approach successfully focuses the difficult learning task (compositional blending) onto a small, dedicated model (the IP-Prior) while leveraging the decoding power of a much larger, pre-trained model.

**Weaknesses:**

1. An important step in validating the PartEval metric is missing. The paper does not report running PartEval against the HiDream-11 ground-truth images used for training. This omits the essential check required to verify that the Gemini pipeline accurately assigns a perfect score to the ideal compositional result (which is being used as ground truth during the training).
2. SAMv2 obtained masks play a critical role in the practical utility of the proposed method. The authors should perform analysis demonstrating the IP-Prior's robustness to segmentation errors.
3. The training structure suggests the model may be relying on memorized structural templates (slots for head, body, wheels, etc.) derived from the limited taxonomy, rather than learning general blending logic. The authors should potentially discuss its ability to extrapolate to novel or conflicting spatial configurations in order to prove the robustness of the approach.
4. In the fig 6. qualitative results., some results aren't that good: e.g, in the first column, the generated image does seem to have impact from zebra's body (see the thicker black tail) and not just zebra's head. I would appreciate if the authors can provide an explanation to the same.

**Questions:**

1. I wonder how reliable HiDream-11 generations would be? For the examples shown in figure 3, for the prompt mentioning a plant with petals of rose and leaves of sunflower the generated image also has the yellow petals from sunflower which may not be desirable.
2. Is the IP-prior sensitive to perturbations/minor errors in the SAMv2 masks?
3. I would appreciate if the authors can provide some intuition for how IP-prior would resolve potential spatial conflicts in a potential such scenarios when you have multiple input parts to be used in the composition. For instance, if the input had a cow and a horse and the user wants both of them to play a role in how the head of the compositionally generated object looks like, would the model perform some blending, or pick one of the two?

---

> ### Author Response · Authors · 2025-11-20
>
> Thank you for your time and your review of our work. Below, we address your questions and the weaknesses mentioned.
>
> **Validating PartEval metric with Hi-Dream generated dataset:** We acknowledge the importance of validating PartEval against the Hi-Dream ground-truth images. However, since Hi-Dream is a text-to-image  model, it does not directly align with the input requirements of PartEval, which expects both image inputs and text annotations for evaluation. Our current setup focuses on distilling knowledge from Hi-Dream, a large, well-trained model, into a smaller model to ensure it learns effective part-level compositionality to produce novel objects.
> While we conducted experiments comparing PartEval scores with human evaluations and observed notable differences (as shown in our line graph analysis), this highlights that PartEval does not yet fully match human judgment. We consider PartEval a preliminary step toward an automated, interpretable evaluation of compositional image generation, and we hope future work will build upon and refine this direction.
>
> **Robustness of IP-Prior with SAMv2 errors:** Thank you for the suggestion. We will include additional experiments to analyze the performance and robustness of IP-Prior under segmentation inaccuracies introduced by SAMv2.
>
> **Resolving spatial conflicts during multi-object concept blending:** Thank you for the suggestion. Based on our preliminary observations, when provided with inputs such as the faces of a leopard and a zebra, the model tends to blend features, capturing the leopard’s facial texture (e.g., spots) while preserving the zebra’s structural attributes (e.g., mouth shape). We will include additional experiments to show how IP-Prior handles multiple input parts in compositional scenarios.

---

### Official Review · Reviewer_uVkz · 2025-10-31

**Soundness:** 3
**Presentation:** 2
**Contribution:** 2
**Rating:** 4
**Confidence:** 4

**Summary:**

The paper proposes Chimera — a personalized image generation model that creates novel hybrid objects by combining parts from different input images. The main contribution of this paper is designing a part taxonomy that contains 464 unique <part, subject> semantic atoms. This taxonomy enables the generation of a large amount of part combination data. However, the paper lacks innovation in aspects such as model structure and experimental setup.

**Strengths:**

- The proposed ⟨part, subject⟩ taxonomy covers 6 semantic domains and 464 semantic atoms, avoiding the limitations of baselines (e.g., PartCraft is restricted to "birds/dogs," and PiT focuses on "toy creatures").
- The paper proposes a new metric that leverages MLLMs to evaluate the part-influenced generation.

**Weaknesses:**

- The second innovation claimed in the paper—"The model does not require masks"—is meaningless. This is because the authors utilize Grounded SAM for image segmentation, which offers no novelty whatsoever.
- The paper provides insufficiently clear descriptions of the training details.
- From a qualitative results perspective, the consistency between the parts of the images generated by the model and the input parts is not good.

**Questions:**

- During model training, are the input part images segmented from ground truth images using Grounded SAM? If so, how to ensure the model's generalization ability?
- Qualitatively, the generated images show very weak adherence to the input parts and text prompts in the 4-part compositions setting. This is particularly evident in the numerous cases presented in Figure 11, where the generated content barely captures the input features. A further explanation on this issue is expected.

---

> ### Author Response · Authors · 2025-11-20
>
> Thank you for your time and your review of our work. Below, we address your questions and the weaknesses mentioned.
>
> **The claim "does not require masks” offers no novelty:**  Since SAMv2 is integrated into the generation pipeline, users are not required to manually provide masks, which helps reduce user effort. This offers practical advantages over previous approaches that rely on explicit user-provided annotations or masks. We will clarify this point in the paper.
>
> **Insufficient training details:**  We train the part-conditioned prior while keeping the decoder frozen for 250K steps on a single NVIDIA A100 GPU with a batch size of 64, using the Adam optimizer (β₁=0.9, β₂=0.999) and a learning rate of 1×10⁻⁵. The training and validation datasets are constructed from Hi-Dream generated images, consisting of paired source and target samples. Source images are part-level crops automatically extracted from the target images using Grounded SAM v2, guided by predefined keywords (e.g., head, body, tail, legs) for each object class. The model is then trained to reconstruct the original images from these part-level inputs.  Each sample includes up to five cropped parts and a 2×2 grid of related images from the same category. Random augmentations such as rotation, cropping, and sketch perturbations (applied with probability 0.5) are used to improve robustness. These details will be described in the paper.

---

### Official Review · Reviewer_JDyS · 2025-11-01

**Soundness:** 2
**Presentation:** 2
**Contribution:** 2
**Rating:** 2
**Confidence:** 4

**Summary:**

This paper proposes a personalized image generation model that generates new objects by combining specified parts from different
source images according to textual instructions. A training dataset is built with <part, subject> pairs. A metric PartEval pipeline is introducted to assess the fidelity and compositional accuracy.

**Strengths:**

- The task of compositional generatation of new objects by combining specified parts from different source images according to textual instructions is investigated.
- The experiments are reported and analysis.

**Weaknesses:**

- Qualitative results comparisons is insufficient.
- Failure cases are not provided.
- The number of compositional objects is small, restricting generalization to complex, real-world scenes with diverse element combinations.
- The code is not provided,  hindering reproducibility and further validation of the proposed approach.

**Questions:**

See above.

---

> ### Author Response · Authors · 2025-11-20
>
> Thank you for your time and your review of our work. Below, we address your questions and the weaknesses mentioned.
>
> **Qualitative results comparisons are insufficient:**  We will include additional results in the revised version.
>
> **Failure cases are not provided:**   We will add representative failure cases in the appendix.
>
> **The number of compositional objects are limited:** Our dataset includes diverse categories (creatures, vehicles, plants, furniture etc.)  with varying levels of complexity. We present results demonstrating broad generalization across 2, 3,  and 4-part compositions. We would be happy to address any specific concerns for further clarification.
>
> **The code is not provided:** The code is already provided in the anonymous Github repository linked in the paper.

---

### Author Response · Authors · 2025-11-20

We thank the reviewers (R1: JDyS, R2: uVkz, R3: aNSa, R4: RB71) for their thoughtful feedback and for recognizing the significance of addressing the problem statement (R1, R4), the contribution of our extensive data taxonomy (R2), the introduction of a new metric for evaluating part-based generation (R2, R3), and the proposed method for automatically creating hybrid compositions without user-provided masks (R3). We will address the reviewers’ concerns in the revised version and are currently awaiting the results of additional experiments.  Below, we respond to some general concerns and outline the planned revisions.

**R2, R3, R4:  The qualitative results are not consistent enough with inputs:**  The observed inconsistency arises because, while the model explicitly incorporates the provided parts, it also exercises generative freedom for unspecified regions, leading to some stochastic variation in outputs. For example, if given the head of a zebra and the body of a gazelle, the model may sometimes include a zebra’s tail along with the head  since the specifications for the tail were not specified. To address this, we will conduct additional inference experiments using a larger decoder (while keeping the IP-Prior fixed), the current setup uses an SDXL decoder, and include the corresponding qualitative results.


**R3, R4: Reliability of Hi-Dream generated data:** We would like to clarify that our synthetic dataset provides reliable supervision for part-based learning. Since our prompts only ask for specific parts (e.g., rose petals, sunflower leaves), the  noted "artifacts" (such as the yellow outer petals in Figure 3)  reflect Hi-Dream’s completion behavior, filling missing context to maintain visual coherence, connecting details like the outer petals so the image does not look like a disjointed collage.  As long as the specific request parts are correct, the model learns the right concepts.

---

### Meta-Review · Area_Chair_rkJL · 2026-01-07

**Summary:**

This paper proposes a part-based approach for compositional image generation. Reviewers proposed various concerns, including novelty of the proposed approach, performance validation, result comparisons, identity preservation, etc. This paper receives three negative scores and 1 positive score pre-rebuttal.

**Reviewer Concerns:**

The authors answered some of the reviewers' questions, but not all. In addition, some answers are not convincing, only mentioning that authors will study the question in future versions.

**Reviewer Scores:**

Since the authors did not answer the reviewers' questions thoroughly, it is unlikely that the negative reviews will increase their scores after rebuttal.

---

### Decision · Program_Chairs · 2026-01-26

Reject